# Characterization of Vaccine Breakthrough Cases during Measles Outbreaks in Milan and Surrounding Areas, Italy, 2017–2021

**DOI:** 10.3390/v14051068

**Published:** 2022-05-17

**Authors:** Silvia Bianchi, Maria Gori, Clara Fappani, Giulia Ciceri, Marta Canuti, Daniela Colzani, Marco Dura, Mara Terraneo, Anna Lamberti, Melissa Baggieri, Sabrina Senatore, Marino Faccini, Fabio Magurano, Elisabetta Tanzi, Antonella Amendola

**Affiliations:** 1Department of Health Sciences, Università Degli Studi di Milano, 20142 Milan, Italy; silvia.bianchi@unimi.it (S.B.); maria.gori@unimi.it (M.G.); clara.fappani@unimi.it (C.F.); giulia.ciceri@gmail.com (G.C.); marta.canuti@gmail.com (M.C.); daniela.colzani@unimi.it (D.C.); marco.dura@outlook.it (M.D.); mara.terraneo@unimi.it (M.T.); elisabetta.tanzi@unimi.it (E.T.); 2Department of Clinical Sciences and Community Health, Università Degli Studi di Milano, 20133 Milan, Italy; 3Health Protection Agency, Metropolitan Area of Milan, 20121 Milan, Italy; alamberti@ats-milano.it (A.L.); ssenatore@ats-milano.it (S.S.); mfaccini@ats-milano.it (M.F.); 4Department of Infectious Diseases, Istituto Superiore di Sanità, 00161 Rome, Italy; melissa.baggieri@iss.it (M.B.); fabio.magurano@iss.it (F.M.); 5Coordinated Research Center “EpiSoMI”, Università Degli Studi di Milano, 20133 Milan, Italy

**Keywords:** measles virus, measles vaccine, vaccination failure, hemagglutinin

## Abstract

Despite the existence of an effective live-attenuated vaccine, measles can appear in vaccinated individuals. We investigated breakthrough measles cases identified during our surveillance activities within the measles/rubella surveillance network (MoRoNet) in Milan and surrounding areas (Northern Italy). Between 2017 and 2021, we confirmed measles virus (genotypes B3 or D8) infections in 653 patients and 51 of these (7.8%) were vaccinees. Among vaccinated individuals whose serum was available, a secondary failure was evidenced in 69.4% (25/36) of cases while 11 patients (30.6%) were non-responders. Non-responders were more frequently hospitalized and had significantly lower Ct values in both respiratory and urine samples. Median age and time since the last immunization were similar in the two groups. Importantly, we identified onward transmissions from vaccine failure cases. Vaccinees were involved in 20 outbreaks, in 10 of them they were able to transmit the virus, and in 8 of them, they were the index case. Comparing viral hemagglutinin sequences from vaccinated and non-vaccinated subjects did not show a specific mutation pattern. These results suggest that vaccination failure was likely due to the poor immune response of single individuals and highlights the importance of identifying breakthrough cases and characterizing their clinical and virologic profiles.

## 1. Introduction

Measles is the most transmissible human infectious disease and, although an effective vaccine is available, it remains a major cause of death in children [1]. For this reason, all six World Health Organization (WHO) regions have established or expressed a commitment to achieving regional control or elimination of measles, although the targets and milestones on the path to elimination vary between regions [2].

Italy is one of the 16 countries in the European region where measles is still endemic [3,4]. The measles vaccine has been recommended here since 1979, but a low vaccination coverage was reported in the 1980s and 1990s. Although the vaccination rate improved in subsequent years, it never reached the 95% target required for elimination [5]. Currently, the administration of two doses of the measles-mumps-rubella (MMR) vaccine, the first dose at 12–15 months and the second dose at 5–6 years of age, is the recommended schedule [6] and, since July 2017, the MMR vaccine became mandatory for all children up to the age of 16 years [7]. Although there are 24 recognized measles virus (MV) genotypes [8], the MV is considered serologically monotypic [9] and most of the available vaccine strains derive from a strain isolated in 1954 [10].

Outbreaks involving more than 10,000 cases have been reported in Italy from 2017 to 2019. Since late 2019, a consistent decrease in measles cases has been observed, with only 103 cases reported in 2020 and 3 cases observed between January and August 2021. This reduction is most likely due to the interruption of surveillance activities because of COVID-19-related lockdown measures and the behaviors adopted in response to the SARS-CoV-2 spread [11,12]. Large measles outbreaks in the years immediately preceding the COVID-19 pandemic were also reported in Milan and surrounding areas [13]. As the Subnational Reference Laboratory (SRL) of the Italian measles and rubella surveillance laboratories network MoRoNet in Milan, we investigated over 800 measles cases in 5 years, some of which occurred in previously immunized individuals and could be associated with vaccination failure. Two different mechanisms that could be responsible for vaccine failure are described [14]. Primary vaccination failure is the absence of a humoral immune response (lack of IgG production) [14,15]. Subjects that fail to seroconvert after the vaccination are also referred to as non-responders. Secondary vaccination failure consists of a sub-optimal or non-protective response to immunization by the vaccination or in the loss of vaccine-induced immunity over time [14,16,17]. In this case, an infection can occur even if subjects present IgG in their blood.

Here, we present a detailed analysis of breakthrough measles cases observed between 2017 and 2021 in Milan and surrounding areas (Northern Italy). The aim of the study was to characterize the serological and clinical profiles of cases of vaccination failure. In addition, since neutralizing antibodies and the humoral immune response are mainly directed against the hemagglutinin (H) protein [18], a phylogenetic analysis of the H gene was performed to compare strains identified in vaccinated and non-vaccinated cases.

## 2. Materials and Methods

### 2.1. Study Population

This study included 864 suspected measles cases (430 females, 434 males) observed between March 2017 and December 2021 in the metropolitan area of Milan and its surroundings, a densely populated area of about 4 million inhabitants, as well as other Lombardy provinces, i.e., Brescia, Varese, Como, and Monza-Brianza. The median age of the investigated cases was 29 years (range: 0–89 years). A total of 742 cases (85.9%) were unvaccinated individuals.

Samples were collected as part of the measles and rubella surveillance framework and included urine, oropharyngeal swabs, and sera.

### 2.2. Molecular and Serological Methods

Measles RNA was detected in urine and/or oropharyngeal swabs (Copan, Copan Italia SPA, Brescia, Italy) submerged in a universal transport medium with a multiplex One-Step Real-time RT-PCR (rRT-PCR) targeting the nucleoprotein (N) gene of the MV and the P150 gene of the rubella virus, as previously described [19]. MV genotyping was performed by sequencing the highly variable region of the nucleoprotein gene (N-450), as recommended by WHO [20], after a hemi-nested PCR [21,22]. The H gene of a selection of strains has been characterized as previously reported [23].

Anti-measles antibodies were detected in serum samples collected during the acute phase of infection (1–10 days after symptoms onset) by semi-quantitative (for IgM detection) and quantitative (for IgG detection) ELISA performed with the Anti-Measles Virus ELISA IgM/IgG kit (Euroimmun, Lübeck, Germany). Briefly, according to the manufacturer’s specifications, IgM results were interpreted as follows: negative <0.8, 0.8 ≤ equivocal < 1.1, and positive ≥1.1; the levels of IgG were indicated in International Units per liter (mIU/mL) and evaluated as follows: <200 mIU/mL—negative; 200–275 mIU/mL—borderline; and ≥275 mIU/mL—positive. Vaccine failures were classified according to the presence of IgG and IgM in these samples into primary (IgM−/IgG−, IgM+/IgG−) or secondary (IgM−/IgG+, IgM+/IgG+) [24]. Sera of cases with a definition of a secondary vaccine failure were confirmed through an antibody avidity assay using the Anti-Measles Virus ELISA IgG Avidity kit (Euroimmun, Lübeck, Germany), following the instructions provided in the manual.

### 2.3. Statistical Analysis

Comparisons of proportions or means were accomplished by the Chi-square test and *t*-test, respectively. Two-sided *p* values < 0.05 were considered statistically significant. All analyses were conducted using the OpenEpi online tool [25].

### 2.4. Phylogenetic Analysis of H Gene Sequences

Sequences obtained in this study were compared to those previously identified from the same monitored area [23,26], as well as to reference sequences from the B3-400V and D8-400T clades [23] obtained from the GenBank database. Sequence alignments performed with Clustal X [27] were used to build maximum likelihood phylogenetic trees with MEGA 11 [28] using the best substitution model identified by a model test. Branch robustness was evaluated by bootstrapping (100 replicates) [29]. Sequences included in this study have been deposited in the MeaNS database [30].

## 3. Results

### 3.1. Measles Epidemiology in the Surveilled Area

Out of 864 cases investigated between 2017 and 2021, MV infection was confirmed in 653 patients (75.6%). A total of 51 measles cases (7.8%, median age of 22.5 years) were individuals who previously received one (26 cases, 4.0%) or two (25 cases, 3.8%) MV vaccine doses. During the 2017 epidemic, 6.9% (23/334) of confirmed measles cases had a history of vaccination. This proportion was 9.6% (10/104) in 2018 and 8.5% (18/211) in 2019. None of the 2020–2021 confirmed measles cases (N = 4) were in vaccinated subjects (Figure 1).

Out of the 51 cases that occurred in vaccinated individuals, 26 (51.0%) were defined as sporadic while the other 25 were related to 20 different outbreaks. Interestingly, transmission from ten vaccinated individuals was reported. Specifically, vaccinees gave rise to eight different familiar or nosocomial outbreaks involving two to four people (Table 1). The eight index cases were all females with a median age of 27 years.

A total of 18 out of 51 (35.3%) vaccinated cases received their dose after the age indicated in the National Vaccination Program. On average, those who received only one dose were vaccinated 13.4 years after the recommended age for the first dose (13–15 months), while those who received two doses were vaccinated 6.1 years after the recommended age for the second dose (5–6 years) (difference not significant, ns).

Viral N-450 sequences were obtained from 36 of these vaccinated cases, as well as from other 562 cases of unvaccinated infected subjects. D8 and B3 were the only genotypes identified. Related cases and outbreaks are described in Table 1.

### 3.2. Serological and Clinical Profiles of Breakthrough Cases

For a subgroup of 38 (74.5%) of the 51 vaccinated cases, a serum sample collected during the acute phase was available; for 36 of these, it was possible to distinguish between primary and secondary vaccine failures (Figure 2).

A secondary failure was initially evidenced in 27 cases and all these subjects presented IgG titers above protective levels (>120 mUI/mL [31]) in their sera (>580 mUI/mL for IgM−/IgG+ patients and >760 mUI/mL for IgM+/IgG+ patients). The subsequent IgG avidity testing demonstrated high avidity antibodies (relative avidity index >60%) for 25 out of 27 samples. For one sample, equivocal results were obtained (relative avidity index between 40% and 60%), and for another sample, the result was out of the range of valid values (optical density <0.140). This test, therefore, confirmed secondary vaccination failure in 25/36 (69.4%) subjects for which a distinction between primary and secondary vaccine failure could be made. In both primary and secondary failure groups, almost half of the subjects received two doses of vaccine. The characteristics of individuals with primary and secondary failure are reported in Table 2.

Onward transmission was recorded for ten vaccinated subjects, including four cases of primary vaccination failure, four cases of secondary failure, and two subjects whose sera were unavailable for antibody testing (Table 2). Of these ten, eight were index cases (Table 1). Overall, higher evidence of transmission was observed in the non-responder group (36.3% vs. 16%, *p* = 0.2179, ns).

Out of the 653 confirmed measles cases, 604 (93.4%) were reported by hospitals, 33 of which were vaccinated subjects (5.5%). A higher proportion of measles cases with primary vaccination failure (10/11, 90.9%) needed hospital care compared to cases of secondary failure (19/25: 76%, *p* = 0.3485). Finally, attendance at a hospital emergency room was significantly associated with no history of vaccination (95.2% vs. 70.2% for unvaccinated and vaccinated individuals, respectively; *p* < 0.01).

Real-time RT-PCR was carried out on a total of 373 oropharyngeal swabs and 564 urine samples. A total of 42 oropharyngeal swabs and 46 urine samples collected from patients with a history of vaccination showed a significantly higher mean cut-off point of cycle threshold (Ct) values compared to those with no vaccination history (Table 3). The Ct values obtained from both oropharyngeal swabs and urine samples were lower for non-responder patients than for patients with secondary vaccination failure (Table 3).

### 3.3. Phylogenetic Analysis of the H Gene

We compared the H gene and protein sequences obtained from vaccinated and unvaccinated individuals. H gene sequences were investigated for 79 subjects that were identified to be measles-positive in Lombardy between 2017 and 2019, including 24 sequences from subjects that previously received one or two measles vaccination doses.

Phylogenetic trees built on these sequences showed that, for both B3 and D8 genotypes, sequences from Lombardy could be included in various clades, independently from the vaccination status (Figure 3 and Figure 4). These corresponded to the various genetic strains circulating in Lombardy in the different years, as we also showed previously [23,26]. Additionally, comparing the amino acid sequences of strains within each clade did not result in the identification of specific mutations that could be associated to immune escaping, considering both non-responders and cases of secondary vaccination failure. In fact, in most cases, sequences from vaccinees were identical or nearly identical to sequences from non-vaccinated subjects and we could not identify a specific mutation pattern in viruses identified in vaccinees. Finally, only 1 (247 L) of the identified mutations was included in one of the five immune epitopes [23].

## 4. Discussion

Despite the availability of an effective and safe live attenuated vaccine, measles can appear in vaccinated individuals. Indeed, in a review, Haralambieva et al. [34] described the lack of protective antibody titers in 2–10% of vaccinated individuals and the existence of non-responders to anti-measles vaccines (≤2.6% after two doses) [35].

In Italy, even if vaccine coverage has increased after the introduction of the mandatory vaccination in 2017, the optimal target for measles elimination has not yet been achieved. During the last outbreaks that occurred in Milan, we observed that the majority of cases involved unvaccinated individuals as the proportion of measles breakthrough cases was as low as 7.8%. This result is in line with national data but lower compared to those countries where vaccination rates are high and measles endemic transmission has been interrupted, and where a higher proportion of cases occur in susceptible vaccinated subjects [36]. Indeed, the frequency of measles cases among vaccinees increases as the vaccination rate increases in the general population [37]. Our results are consistent with the assumption that a sub-optimal vaccination coverage leads to the development of pockets of susceptible individuals in the population which favors the occurrence of outbreaks that could also involve vaccinated individuals that did not develop protective immunity. This highlights the need for catch-up vaccination campaigns targeting older, unvaccinated individuals to increase the overall vaccination coverage and reduce the chance of outbreaks.

In our study, we identified cases of both primary and secondary vaccination failure. Even if most of the breakthrough vaccine cases (69.4%) could be classified as secondary failure, a higher percentage of hospitalization and lower Ct values, both in respiratory material and in urine, were observed in non-responders. Although a precise viral load measurement was not performed, these data are consistent with other studies that recognized a milder disease [38] and a lower rate of transmission in secondary failures. There are only a few cases of measles transmission from vaccinated individuals described in the literature and, most of the time, these occurred after prolonged familiar and nosocomial close contacts and involved primary vaccination failures [39,40]. Importantly, we report onward transmissions from primary and secondary vaccine failure cases, with vaccinated people acting both as index cases and as secondary transmitters. Indeed, vaccinees were involved in 20 outbreaks (mainly familiar or nosocomial) in the studied area, and in 50% of these, it was demonstrated that vaccinated subjects were able to transmit the virus. In 40% of the cases, vaccine receivers were the index case and half of these subjects received two vaccination doses. This emphasizes that all MV infected subjects, including vaccinated individuals, can transmit the virus and should carefully follow virus containment procedures. This is of particular importance in health care settings.

The type of vaccine failure did not correlate to the number of vaccine doses since half of both non-responders and secondary failure cases received two vaccine doses. However, these values could have been affected by the small sample sizes of both vaccine failure groups.

In the five years of measles laboratory surveillance, only the two globally endemic genotypes B3 and D8 were observed in the area under surveillance. This seems to be in line with literature data, as these two genotypes are currently causing large outbreaks worldwide [30]. Most of the strains identified in vaccinated subjects were classified as genotype D8, but this was probably due to the fact that D8 was the dominant genotype in Italy during the 2017 and 2019 epidemics [21,41]. Phylogenetic and sequence analysis of the hemagglutinin gene showed that similar strains were circulating in MV-positive individuals independently of the vaccination status, and specific mutations associated with immune escaping were not identified. Our results suggest that vaccination failure is, therefore, not associated with the circulation of immune escaping mutant strains but, more likely, the breakthrough is due to the poor immune response of single individuals.

## 5. Conclusions

The last measles-positive specimen tested by our SRL was in February 2020. Likewise, in May 2020, the Union/European Economic Area countries reported the lowest number of measles cases since January 2010 [12]. Nonetheless, because of the ongoing COVID-19 pandemic, the risk of measles outbreaks is rising worldwide. Indeed, all six WHO Regions have reported a disruption in immunization activities, with major consequences on both routine immunization and mass vaccination campaigns [42]. Moreover, the WHO estimates that the delays in vaccination campaigns have led to a million children missing scheduled measles vaccine doses [43], leaving cohorts of children unprotected and causing an accumulation of susceptible individuals that could lead to new measles outbreaks. Containment measures implemented against the COVID-19 pandemic are significantly reducing MV circulation. This gives us the opportunity to accelerate progress towards measles elimination [42] by strengthening vaccination programs and by maintaining high levels of surveillance activity to promptly identify new introductions of the virus [44].

The Measles and Rubella Strategic Framework 2021–2030 underlines how the case-based surveillance of measles, together with high-quality laboratory surveillance for timely detection of cases/outbreaks, is important for defining susceptible people and populations and for the implementation of control measures, including the development of adequate immunization policies and strategies [2]. In line with these recommendations, our data show the importance of measles surveillance not only for tracking measles outbreaks but also to identify breakthrough cases and characterize their clinical and virological profiles.

## Figures and Tables

**Figure 1 viruses-14-01068-f001:**
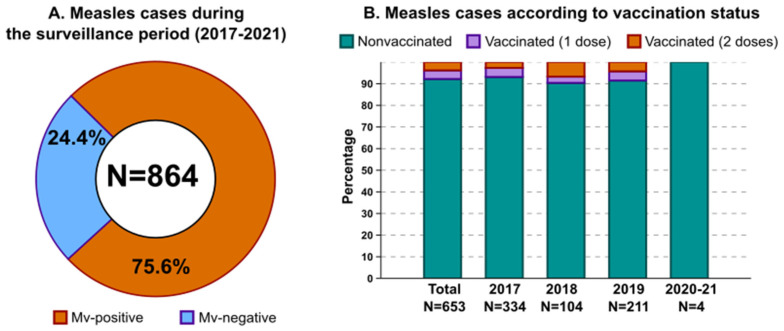
Measles cases in Milan and its surroundings (Northern Italy) during the years 2017–2021. (**A**) The proportion of measles cases identified during the study period. (**B**) The proportion of measles cases according to vaccination status and year of sample collection.

**Figure 2 viruses-14-01068-f002:**
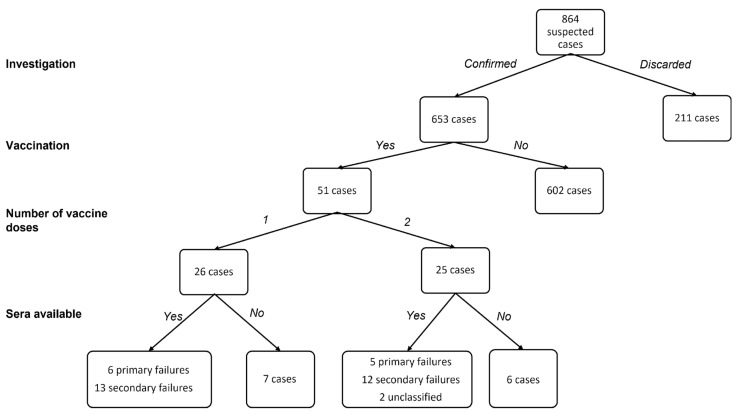
Patient inclusion flow-chart.

**Figure 3 viruses-14-01068-f003:**
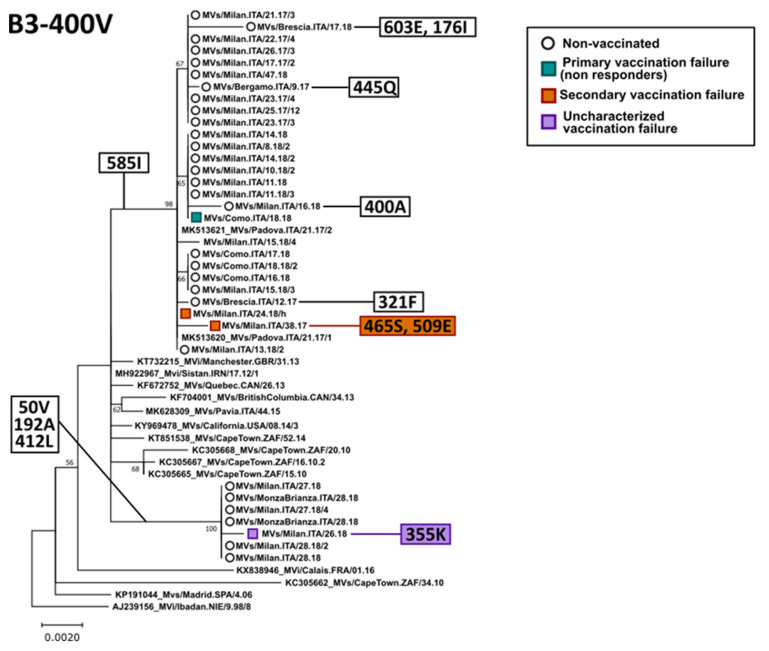
Phylogenetic analysis of genotype B3 H nucleotide sequences belonging to clade B3-400V. The tree was obtained with the maximum-likelihood method [32], based on the Kimura 2 parameters model [33], identified as the best-fitting model after the model test analysis, using MEGA 11 [28]. A discrete Gamma distribution was used to model the evolutionary rate differences among sites (+G = 0.1) and branch support (100 bootstrap iterations [29]) is provided next to nodes. The tree was built using full-length sequences and the phylogenetic placement of a few partial sequences was manually placed on the tree based on a separate phylogenetic analysis with partial H genes [23]. Strain MVi/Ibdan.NIE/9.98/8 was used as an outgroup. Sequences from Lombardy are indicated by an empty circle for non-vaccinated individuals or a square shape for vaccinees (green: primary vaccination failure, orange: secondary vaccination failure, and purple: uncharacterized vaccination failure). Amino acid mutations associated with branches, including studied strains, are indicated in white boxes on the left, while those identified in single strains are indicated on the right by white, green, or orange boxes for non-vaccinated, primary, and secondary vaccination failure, respectively.

**Figure 4 viruses-14-01068-f004:**
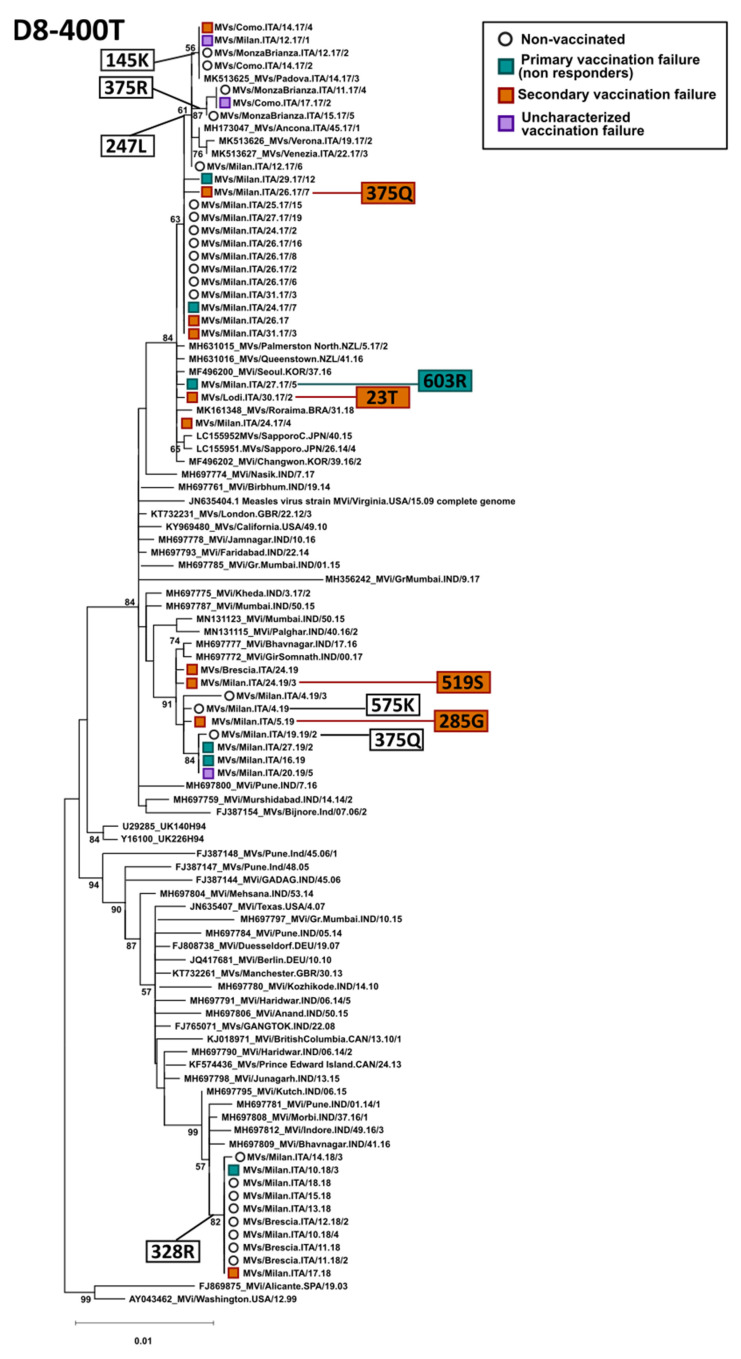
Phylogenetic analysis of genotype D8 H nucleotide sequences belonging to clade D8-400T. The tree was obtained with the maximum-likelihood method [32], based on the Kimura 2 parameters model [33], identified as the best-fitting model after the model test analysis, using MEGA 11 [28]. A discrete Gamma distribution was used to model the evolutionary rate differences among sites (+G = 0.2054) and branch support (100 bootstrap iterations [29]) is provided next to nodes. The tree was built using only full-length sequences and the phylogenetic placement of a few partial sequences was manually placed on the tree based on a separate phylogenetic analysis with partial H genes [23]. Other D8 sequences (MVi/Alicante.SPA/19.03 and MVi/Washington.USA/12.99) were used as an outgroup. Sequences from Lombardy are indicated by an empty circle for non-vaccinated individuals or a square shape for vaccinees (green: primary vaccination failure, orange: secondary vaccination failure, and purple: uncharacterized vaccination failure). Amino acid mutations associated with branches including, studied strains, are indicated in white boxes on the left, while those identified in single strains are indicated on the right by white, green, or orange boxes for non-vaccinated, primary, and secondary vaccination failure, respectively.

**Table 1 viruses-14-01068-t001:** Measles cases and their vaccination history in the metropolitan Milan area and its surroundings during the years 2017–2021.

		Breakthrough Cases					
Genotype	Total Cases	Cases		Outbreaks			
		Vaccinated (%)	Sporadic Vaccinated	N Outbreaks	Index Cases	Subjects Involved (Vaccinated; Not Vaccinated)	Mean Outbreak Size
D8	438	28 (6.4)	8	15	5	20; 31	3.4
B3	160	8 (5.0)	3	5	3	5; 12	3.4
NT ^1^	55	15 (27.3)	15	0	0	0	0
Total	653	51 (7.8)	26	20	8	25; 43	3.4

^1^ NT: Not Typed.

**Table 2 viruses-14-01068-t002:** Characteristics of individuals with primary and secondary MV vaccine failure.

		Median Age, Years (Range)	Time from Rash Onset to Last Vaccine Dose, Average Years (Range)	Mean Age at Last Vaccine Dose, Years (Range)	Evidence of Onward Transmission, N (%)
Primary failure (N = 11) ^1^	Two vaccine doses (N = 5)	27 (6–34)	17 (5–31)	9.2 (5–12)	2 (40.0)
	One vaccine dose (N = 6)	29 (2–43)	18 (1–27)	8.7 (1–33)	2 (33.3)
Secondary failure (N = 25) ^2^	Two vaccine doses (N = 12)	18 (10–27)	12.3 (5–22)	7.6 (5–11)	1 (8.3)
	One vaccine dose (N = 13)	26 (2–38)	20.2 (1–37)	2.6 (1–8)	3 (23.1)
Unclassified (N = 15) ^3^	Two vaccine doses (N = 8)	19 (9–31)	12 (1–31)	7.8 (5–16)	1 (12.5)
	One vaccine dose (N = 7)	26 (3–35)	18 (0–33)	6.9 (1–37)	1 (14.3)

^1^ IgM−/IgG−, IgM+/IgG−; ^2^ IgM−/IgG+, IgM+/IgG+; ^3^ No available serum or vaccine failure classification not confirmed by IgG avidity test.

**Table 3 viruses-14-01068-t003:** Mean cut-off point of cycle threshold (Ct) value comparison between vaccinated and unvaccinated cases and patients with primary or secondary vaccination failure.

	Oropharyngeal Swab			Urine		
	N	Ct Value (Mean)	*p* Value	N	Ct Value (Mean)	*p* Value
Patients with vaccination history	42	30.71	0.0001774	46	31.96	0.000000150
Patients with no vaccination history	331	27.34		518	27.21	
Patients with primary vaccination failure	9	26.38	0.01068	11	28.34	0.00004798
Patients with secondary vaccination failure	25	32.58		25	34.36	

## Data Availability

The sequences generated in this study have been submitted to the WHO’s MeaNS (Measles Nucleotide Surveillance) database.

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
