# Peer review of "Characterization of Vaccine Breakthrough Cases during Measles Outbreaks in Milan and Surrounding Areas, Italy, 2017–2021"

_viruses, 2022, doi:10.3390/v14051068_

Round 1
Reviewer 1 Report
The authors have improved the manuscript by clarifying points and making additions to the text.
Author Response
We thank the Reviewer for the positive comments on our manuscript.

Reviewer 2 Report
I have reviewed the amended manuscript containing the author's reactions to reviewer's comments and found it sufficiently improved in order to recommend its publication in the Viruses journal.
Author Response
We appreciate the Reviewer for the positive feedback.

Reviewer 3 Report
The reviewer appreciates that the proper avidity testing has been added. There is still a lack of methods for the quantitation of measles IgG by ELISA. Again if the measles 3rd standard was used this is not recommended for ELISA based testing, but the method of quantitation needs to be defined, "according to the manufacturers recommendations" is not sufficient. Additionally, the titers obtained by this method would not be in agreement with expected titers of a secondary vaccine failure case, which would be expected to be substantially higher, so this method for determining titer should be clearly described and ensured to be accurate.
The reviewer appreciates the reference of the multiplex real-time PCR, however, still there is no indication of a house-keeping control being used.
I think it would be appropriate to note that within the group you are characterizing as secondary vaccine failure, half of those individuals are considered under vaccinated as they only received one dose of MCV.
Table 1 is still confusing. I appreciate that the authors have expanded table 1, however, the subjects involved column still is unclear.
Author Response
We thank the Reviewer for the accurate reviewing process and the appreciation of our work.
- Following the Reviewer’s suggestion, we have added more details to the ELISA-based testing protocols. As
stated by the manufacturer’s instructions, the “Anti-Measles Virus ELISA (IgG) test” kit by EUROIMMUN (code
EI 2610-9601 G) provides a quantitative or semiquantitative assay. Semiquantitative analyses require an
incubation step with a calibrator (calibrator 3) along with the positive and negative controls and patient
samples. Quantitative analyses require an incubation step with four different calibrators (1 to 4) along with the
positive and negative controls and patient samples. Results can be evaluated semiquantitatively by calculating
a ratio of the extinction value of the control or patient sample over the extinction value of the calibrator. All
samples included in our study were tested with the quantitative protocol and this has now been specified in
the manuscript. Following the Reviewer’s recommendations, we have subsequently performed an IgG avidity
test using a commercially available ELISA test (EUROIMMUN, code EI 2610-9601-1 G) to confirm secondary
vaccine failures. We have updated the Materials and Methods and the Results sections accordingly.
- As we already explained during the previous revision round, our Real-Time PCR protocol does not include
the detection of a housekeeping gene. Since we had to resubmit the article as a new submission, maybe the
Reviewer did not see the responses we provided to his/her previous comments.
- We thank the Reviewer for pointing out this aspect. We completely agree that a proportion of those who
received only one dose of vaccine could have been avoidable by implementing the second dose.
- To address the Reviewer's concern, we have modified Table 1 in the revised manuscript, in order to increase
its clarity.

This manuscript is a resubmission of an earlier submission. The following is a list of the peer review reports and author responses from that submission.
Round 1
Reviewer 1 Report
This paper seeks to understand and characterize vaccine breakthrough cases in Northern Italy occurring between 2017 and 2021. The study includes 864 suspected measles cases, the majority of which were in unvaccinated individuals. Of the confirmed cases 7.8% were in previously vaccinated individuals. Authors used an inhouse real-time method for testing urine or OP swabs, and the genotyping was performed by WHO standard N-450 sequencing. Additionally, the authors sequenced the H gene of selected strains and evaluation of sequence data was performed using standard methods for phylogenetic tree analysis. Along with OP or urine serum was collected during the acute phase of infection and IgM and IgG evaluated by ELISA. Vaccine failures where classified by the presence of IgM and IgG with IgM positive IgG negative cases representing primary vaccine failures and IgG positive cases representing secondary vaccine failures.
There are several instances of incorrect citation of data or literature and it is recommended that these be resolved by the authors. The examples throughout the paper are listed below throughout the comments regarding the manuscript.
In the first paragraph of the introduction, the author states that all 6 WHO regions have set measles elimination goals. Only 5 of the 6 regions have set measles elimination goals, the Africa region has not set an elimination goal, but has set measles control goals. While it may seem a minor discrepancy, this makes the statement as currently phrased incorrect.
The authors report a relatively high transmission rate from vaccine failures but not characterizing if transmission was from single dose or 2-dose vaccinees. This information would be helpful to include and would give better idea of incidence of transmission from secondary vaccine failure cases as previously reported in the literature. While some of that information is captured in table 2 it is not captured for all since table 2 focuses only on those with serum samples available and only 8 cases with evidence of transmission out of the 10 are noted. This information could be included in the text to demonstrate that there was only serological data for vaccine failure in 8 or 10 cases with onward transmission.
Table 1 is unclear to the reviewer. There were 51 breakthrough cases. What then is the number for subjects involved? Additionally, adding a columns to include the total number of outbreak associated and sporadic cases may increase the understanding of this table. With that information if the mean outbreak size calculation may then be more clear to the reader that it is being calculated from the total number of cases associated with an outbreak.
Characterization of secondary vaccine cases would be further supported by understanding the titer of antibody or avidity of antibody in these cases. High avidity antibodies would be useful to serve as a confirmation of secondary vaccine failure and support the current characterization. Without this associated data, even though serum is reported to be collected within the acute phase, the current characterization of secondary vaccine failures is lacking. Most literature uses at a minimum avidity to characterize secondary vaccine failure cases.
The authors indicate that a good proportion of vaccine failure cases received vaccine while older. Given the wide range of age since last dose in both primary and secondary vaccine failure groups and given the small sample size of vaccine failure groups, the reviewer believes that there isn’t enough data to support this statement. Furthermore, in table 2 a secondary vaccine failure with having received two doses of vaccine there is at least one individual that falls into this group with an age of 1 given the range in this line of the table. Given that routine immunization in Italy recommends the first dose at age 12-15 months and the second at 5-6 years, it is unclear how they have a case with 2 doses by age 1. The reviewer then questions whether any of the vaccine failure cases were administered a vaccination dose at the age of 6 months, if so this is considered a 0 dose and should not be included in the two dose group. It is recommended that the authors review the data to ensure that no 6 months doses were being considered as a 1st dose and included in the two-dose group within the study.
While there is currently an interest in looking at the Ct threshold as a measure of viral load, Ct alone is difficult to account for true viral load given that the assay was not calibrated for this. Furthermore, any evaluation of viral load at the very least should be performed by normalizing to the housekeeping gene, as viral RNA input concentration alone can affect Ct and therefore what is being interpreted as viral load. Without at the very least this information this reviewer feels that viral load/Ct differences cannot be used to determine actual viral load differences.
The presentation of the numbers of cases, what was included, excluded, vaccine failure, etc could be clearer. A large comprehensive table or flow chart may be useful.
The methods section should include the primers used or appropriate references for sequencing both for the genotyping as well as the separate H sequencing performed.
Please check the language of sentences, there are a few throughout in which the grammar is incorrect or there is a word missing. For example discussion second paragraph sentence starting “During the last oubreaks [that] occurred in Milan…”
The statement in the discussion that transmission has been documented only in primary vaccine failure cases is incorrect. While the transmission is indeed lower with limited documented transmission from a vaccinated individual, there has been documentation of vaccine transmission from secondary vaccine failures.
Additionally, the statement at the end of the second paragraph of the discussion seems to indicate that it is vaccine failure cases that serve as pockets of susceptible that can give rise to outbreaks. This conclusion this reviewer believes is not fully supported by the data. While there is the potential for limited transmission, the driving force of the outbreaks described in this manuscript was clearly the unvaccinated population. Clearer defined epidemiology would be required in the text of this manuscript regarding the vaccine failure individuals believed to be index cases, and how far transmission within the community occurred from these potential index cases.
While the authors capture the important point that vaccine failure cases have the potential to transmit, it would be good to remind the reader of the proportion of cases overall that were vaccine failures in these discussion points to again focus the reader on the low percentage of vaccine failure cases. Additionally, it was noted in the discussion that the secondary vaccine failure cases were higher but the results section 1st paragraph has nearly equal number of primary and secondary vaccine failure cases in these individuals based on documented doses of vaccine. Because only a proportion of sera was available for further study and characterization the proportion of serum available from documented doses of vaccine would be needed information for the reader. If a lower proportion of available serum samples were used for ELISA testing for those with single dose for vaccine this affects the interpretation of the data. It is recommended that the authors devise a figure to demonstrate available samples as well as onward transmission from vaccinated individuals or revise table 2 to include this information.
In the final paragraph of the discussion. The interpretation that these two genotypes are more transmissible is not supported by the literature. There has been a successive decline in measles genotypes as vaccination use has expanded, and regions with endemic virus have seeded outbreaks in other regions in which measles control efforts have reached greater maturity. Therefore, the diversity of measles genotypes circulating has decreased but there is no evidence that these genotypes are more transmissible or escape protection (which the authors do highlight) mediated through either natural or acquired immunity.
Reviewer 2 Report
I have reviewed with interest the manuscript entitled “Characterization of vaccine breakthrough cases during measles outbreaks in Milan and surrounding areas, Italy, 2017-2021” authored by Silvia Bianchi et al. The paper describes the result of a comprehensive measles surveillance campaign conducted at COVID 19 pre-pandemic times. Its results, although not surprising, documented precisely the high efficiency of the measles vaccine due to the monotypic nature of measles virus and the potential to spread the infection when viremic levels are higher among primary vaccine failures. As such, I found the paper suitable for publication in Viruses, however it needs some data in table 2 and regarding measles virus titers to complement their conclusions, before I can fully recommend its publication. As follows:
- What was the type of measles vaccine used? What was the age at which the first dose was received?
- It would be convenient to have a comparison plot comparing MV neutralizing titers among vaccine failure cases. If the authors were able to distinguish between primary and secondary failure, then it is possible to discern MV titers.
- Authors need to comment on the potential advantage of implementing Measles vaccine campaigns in open population to secure the receipt of two measles vaccine containing doses.
Reviewer 3 Report
The authors have examined the reasons for vaccine breakthrough cases during measles cases in Milan and the surrounding areas between 2017 and 2021. The information obtained is of interest and should inform vaccination policy. The manuscript is generally well written but requires some English grammar editing. There are some minor points which would clarify concepts and help the reader.
- It would be good for the non-specialised reader to actually define non-responder and secondary failure.
- In the materials and methods section 2.2 last line can the authors clarify that the primary failure category is is the same as non-responder even though IgM+, IgG- is in included the latter definition? I think they are different but this is not clear to the reader. In 3.1 page 4 in the results section the authors say they distinguish between primary and secondary failure. Again are non-responders (IgM-, IgG-) included in this? Is there data available for comparison on IgM and IgG levels in vaccinated individuals who do not get infected? Are their IgM and IgG always positive and cell mediated immunity may also be important in protection?
- In Results 3.1. The sentence on individuals who got their last dose later than recommended is a bit confusing. Do the authors mean that the individuals in this category that only received one vaccination got this on average 13.4 years later than recommended. i.e. they would be on average 6 plus 13.4 yrs old (i.e. 19.4 years). Same confusion for those who received a secondary vaccination.
- Results. table 3 and text above. Are these patients a mixture of 1 and 2 dose vaccines? Is there a difference in Ct in relation to doses given?
- Discussion, page 8 2nd paragraph. The 7.8% breakthrough cases being lower than countries where the endemic was interrupted. Can the authors explain further what they mean and particularly what they mean by interrupted.
Round 2
Reviewer 1 Report
The addition of primary and secondary vaccine failure definitions in the introduction are appreciated. It is suggested that the definition of the secondary vaccine failure be adjusted slighted to describe it as sub-optimal or non-protective rather than poor. Most cases of secondary vaccine failure are associated with waning immunity, some have been connected to likely not achieving high enough initial immunity due to vaccine cold chain storage maintenance. The addition of the flow chart and additional data to the figures is appreciated. The flow chart helps for clarity. There remain specific questions regarding the table 1 and additional information in table 2 as outlined below according to the associated response. The majority of the comments of the initial review have been addressed adequately. There remain a couple of comments for further clarification and those are defined below. Additionally, there are two questions that arise from the newly added data, those include regarding the rRT-PCR assay used for the study as newly referenced as well as the addition of the titer of serum determined, both discussed below.
The real-time assay described is a multiplex assay. Was both MeV and RuV run on samples? Additionally, the reference for this assay does not refer to a housekeeping control gene. Was a housekeeping control used in the real-time analysis.
Specific responses to author comments:
Original: In the first paragraph of the introduction, the author states that all 6 WHO regions have set measles elimination goals. Only 5 of the 6 regions have set measles elimination goals, the Africa region has not set an elimination goal, but has set measles control goals. While it may seem a minor discrepancy, this makes the statement as currently phrased incorrect.
Author: We thank the Reviewer for bringing this inaccuracy to our attention. We have rephrased the statement in the revised manuscript as follows: “all six World Health Organization (WHO) regions have established or expressed a commitment to achieving regional elimination of measles and rubella, although the targets and milestones on the path to elimination vary between regions”.
2nd Response: The reviewer appreciates the author trying to adjust this statement. The AFRO region has expressed control goals, not elimination. To make the new statement accurate you still need to adjust the statement. “…commitment to achieving regional [control or] elimination of measles…” And two regions have not set rubella elimination goals.
Original: The authors report a relatively high transmission rate from vaccine failures but not characterizing if transmission was from single dose or 2-dose vaccinees. This information would be helpful to include and would give better idea of incidence of transmission from secondary vaccine failure cases as previously reported in the literature. While some of that information is captured in table 2 it is not captured for all since table 2 focuses only on those with serum samples available and only 8 cases with evidence of transmission out of the 10 are noted. This information could be included in the text to demonstrate that there was only serological data for vaccine failure in 8 or 10 cases with onward transmission.
Author: We thank the Reviewer for the suggestion. Indeed, onward transmission was recorded for 10 subjects, but specific epidemiological/clinical information were previously reported only for subjects with serum samples available. We agree with the reviewer that it is important to provide these data for all patients and we have, therefore, added a row to Table 2 reporting the requested information for the two cases with onward transmission whose sera were unavailable for testing. Moreover, we have added a sentence that explains our results more clearly: “Onward transmission was recorded for ten vaccinated subjects, including four cases of primary vaccination failure, five cases of secondary failure, and two subjects whose sera was unavailable for antibody testing”.
2nd response: The reviewer is not clear why the samples in table 1 do not add up to what is stated in the text. In table 1 the sporadic cases totaled 22, not 26. Outbreak cases listed equaled 24 not 25. And 8 index cases are listed. While the text demonstrates that 2 of these cases were not index cases in that paragraph “Onward transmission was recorded…” the numbers there add up to 11. While it is appreciated that the authors are trying to make this more clear, it is imperative that the numbers add up correctly so that the reader may understand.
Original: Characterization of secondary vaccine cases would be further supported by understanding the titer of antibody or avidity of antibody in these cases. High avidity antibodies would be useful to serve as a confirmation of secondary vaccine failure and support the current characterization. Without this associated data, even though serum is reported to be collected within the acute phase, the current characterization of secondary vaccine failures is lacking. Most literature uses at a minimum avidity to characterize secondary vaccine failure cases.
Author: As the Reviewer pointed out, avidity assays are often used to distinguish between a primary and a secondary immune response and could, therefore, be used to better assess the type of vaccine failure for each case. Unfortunately, we do not have the possibility to perform that test. However, we used a quantitative ELISA to measure IgG levels and we have now included this data in the manuscript. However, we do realize that the lack of avidity measurements is a limitation of our study, and we have, therefore, added a sentence in the conclusions highlighting this possible bias: “In our study we identified both cases of primary and secondary vaccination failure, although the occurrence of a secondary immune response was not confirmed by avidity or neutralization tests”.
2nd Response: The quantitative ELISA is not listed in the methods section. The Euroimmun ELISA does not yield quantitative results. If the authors used an NIBSC standard for quantitation this should also be listed, and if measles standard 3 was used for quantitation, this standard is not recommended for quantitation by ELISA due to variability across results and ELISA kits used when this standard was generated. The standard method to confirm secondary vaccine failure in comparison to primary is still antibody avidity. These tests can be performed with standard a ELISA kit with the addition of a denaturation reagent to test binding strength. This reviewer feels that without this data it makes it hard to make general conclusions such as that the type of vaccine failure did not correlate to the number of vaccine doses.
Original: The authors indicate that a good proportion of vaccine failure cases received vaccine while older. Given the wide range of age since last dose in both primary and secondary vaccine failure groups and given the small sample size of vaccine failure groups, the reviewer believes that there isn’t enough data to support this statement. Furthermore, in table 2 a secondary vaccine failure with having received two doses of vaccine there is at least one individual that falls into this group with an age of 1 given the range in this line of the table. Given that routine immunization in Italy recommends the first dose at age 12-15 months and the second at 5-6 years, it is unclear how they have a case with 2 doses by age 1. The reviewer then questions whether any of the vaccine failure cases were administered a vaccination dose at the age of 6 months, if so this is considered a 0 dose and should not be included in the two dose group. It is recommended that the authors review the data to ensure that no 6 months doses were being considered as a 1st dose and included in the two-dose group within the study.
Author: We thank the Reviewer for pointing out the typo in the age range in Table 2. We have checked the epidemiological investigation form and corrected the table. Furthermore, we agree that the wide range of age since last dose in both primary and secondary vaccine failure groups and the small sample size of vaccine failure groups are a limit of the study. We have adjusted the discussion following the reviewer’s suggestion: “However, despite the wide age range at the time of last dose administration and the small sample size of both vaccine failure groups, non-responders appear to be older at the time of vaccination than subjects characterized by a secondary failure”
2nd response: If the reviewer is reading table 2 correctly then those that were non-responders and secondary failures that received two doses were at the same age at last dose. The difference then the authors choose to focus on in for single dose individuals in which there is significant range of ages at last dose, but the difference in median age could account for differences seen, the median age of those that demonstrated primary failure after a single dose is 6-7 years older than those that demonstrated secondary failure after 1 dose. Typically secondary vaccine failure associated with waning immunity and increases with time since vaccination and looking at time from onset to last dose, that data would be consistent with the literature, and the years since last dose to rash onset is similar in both groups, and in fact would suggest more years of since dose to illness is the primary failure group. While the authors characterize this statement as saying that the results support current recommendations, there is no evidence in the literature that vaccination later leads to less robust response and this study does not meet power to make this type of determination and there is data in the literature that vaccination of adults leads to a protective titer in those that were previously not immune and a boost in titer for those previously immune, albeit temporarily depending on starting titers. The conclusion buried within this concluding statement by the authors is that if you are not immune by adulthood vaccination does not provide sufficient protection, and the literature would argue against this.
Original: Please check the language of sentences, there are a few throughout in which the grammar is incorrect or there is a word missing. For example discussion second paragraph sentence starting “During the last oubreaks [that] occurred in Milan…”
Author: We have carefully checked the grammar and we have made some minor changes in the manuscript.
2nd response: The reviewer sees that the authors have corrected the grammar in many areas. A couple more recommended corrections:
In introduction
“…investigated over 800 measles cases in 5 year, some of which occurred in previously immunized indivuduals and could be associated [with] vaccine failure.” The statement should say associated with not associated to.
“Primary vaccine failure consists [of] the absence…” not consists in. or better yet “primary vaccination failure is the absence of humoral…” Same comments for the secondary vaccine failure statement.
In Results:
“Viral N-450…as well as from [the] other…”